# Associations of BDNF/BDNF-AS SNPs with Depression, Schizophrenia, and Bipolar Disorder

**DOI:** 10.3390/jpm13091395

**Published:** 2023-09-18

**Authors:** Anton Shkundin, Angelos Halaris

**Affiliations:** Department of Psychiatry and Behavioral Neurosciences, Loyola University Chicago Stritch School of Medicine, Loyola University Medical Center, Maywood, IL 60153, USA

**Keywords:** BDNF, BDNF-AS, SNP, depression, schizophrenia, bipolar disorder

## Abstract

Brain-Derived Neurotrophic Factor (BDNF) is crucial for various aspects of neuronal development and function, including synaptic plasticity, neurotransmitter release, and supporting neuronal differentiation, growth, and survival. It is involved in the formation and preservation of dopaminergic, serotonergic, GABAergic, and cholinergic neurons, facilitating efficient stimulus transmission within the synaptic system and contributing to learning, memory, and overall cognition. Furthermore, BDNF demonstrates involvement in neuroinflammation and showcases neuroprotective effects. In contrast, BDNF antisense RNA (BDNF-AS) is linked to the regulation and control of BDNF, facilitating its suppression and contributing to neurotoxicity, apoptosis, and decreased cell viability. This review article aims to comprehensively overview the significance of single nucleotide polymorphisms (SNPs) in BDNF/BDNF-AS genes within psychiatric conditions, with a specific focus on their associations with depression, schizophrenia, and bipolar disorder. The independent influence of each BDNF/BDNF-AS gene variation, as well as the interplay between SNPs and their linkage disequilibrium, environmental factors, including early-life experiences, and interactions with other genes, lead to alterations in brain architecture and function, shaping vulnerability to mental health disorders. The potential translational applications of BDNF/BDNF-AS polymorphism knowledge can revolutionize personalized medicine, predict disease susceptibility, treatment outcomes, and guide the selection of interventions tailored to individual patients.

## 1. Introduction

Brain-Derived Neurotrophic Factor (BDNF) belongs to the neurotrophin family and plays a critical role in promoting the differentiation, growth, and survival of neurons [1,2]. In humans, the BDNF gene is located on chromosome 11p14.1, which is the short arm (p) of chromosome 11 at position 14.1 [3]. It generates multiple transcripts and is expressed in various tissues, with particularly high levels observed in the central nervous system (CNS). It has been shown to influence development, morphology, synaptic plasticity, long-term potentiation (LTP), and brain function [4,5,6,7].

The expression of BDNF is regulated by BDNF mRNA transcription and BDNF protein translation [8], and in the brain, the highest levels are found in the cerebral cortex, hypothalamus, hippocampus, amygdala, medulla, and striatum [4,7,9,10,11]. Synthesis of BDNF protein proceeds in the cell bodies of neurons and glia [5,10]. Its secretion is activity-dependent and can occur from presynaptic and postsynaptic terminals [5]. The BDNF transcript is translated into a pre-pro-BDNF precursor protein, which is further cleaved into the precursor form, pro-BDNF [10,12,13,14,15]. Then, pro-BDNF is cleaved intracellularly into the mature form (m-BDNF), which is then released into the extracellular space and is simply called BDNF [12,13,14,15,16,17].

Cellular membrane depolarization in neuronal cells triggers the release of either pro-BDNF or m-BDNF [18,19]. The proportion of pro-BDNF to m-BDNF fluctuates during specific stages of brain development and across different regions [18]. In the early stages of postnatal development, there is a high concentration of pro-BDNF, while in adulthood, m-BDNF becomes more dominant [15]. However, they both play significant roles in neuronal development, synaptic structure, plasticity, and transmission, although they exhibit distinct functions and engage different receptors, having opposite effects [12,14,20,21,22].

Pro-BDNF preferentially binds to the p75 neurotrophin receptor with its co-receptor, sortilin, and is involved in brain development, long-term depression (LTD), and apoptosis [8,15,21,23]. Moreover, pro-BDNF exerts inhibitory effects on neural stem cell proliferation, differentiation, and migration, leading to a reduction in the number of neurons, oligodendrocytes, astrocytes, and inhibition of neuronal axon outgrowth [15].

In contrast, m-BDNF has a high affinity for the tropomyosin kinase B (TrkB) receptor, promotes neuronal differentiation, and enhances neurite outgrowth, dendritic cell maturation, neuronal survival, synaptogenesis, synaptic plasticity, and connectivity [3,12,17,24].

The expression of BDNF is modulated by a long non-coding RNA gene known as BDNF antisense (BDNF-AS or BDNFOS), which is transcribed from the opposite strand of the BDNF gene [25,26,27,28]. BDNF mRNA and BDNF-AS share a common overlapping region and form double-stranded duplexes, indicating potential interaction [6,25,26].

BDNF-AS serves a regulatory role in controlling BDNF and can suppress BDNF mRNA [26,28,29]. Thus, elevated levels of BDNF-AS were found to cause a reduction in BDNF expression, along with promoting neurotoxicity, an escalation in apoptosis, and a decline in cell viability [30,31]. In contrast, the inhibition of BDNF-AS leads to an upregulation of BDNF mRNA, activating the BDNF-mediated signaling pathway, subsequently increasing BDNF protein levels, suppressing neuronal cell apoptosis, and promoting neuronal outgrowth and differentiation [2,6,28,32].

BDNF can be released in response to various stimuli, including excitatory synaptic activity, hormones, and neuropeptides, highlighting its dynamic regulation and responsiveness to environmental cues [12,33,34]. BDNF plays a significant role in the formation and preservation of dopaminergic, serotonergic, GABAergic, and cholinergic neurons, facilitating efficient stimulus transmission within the synaptic system [18,35,36]. It supports the maturation and stabilization of the cellular and molecular constituents involved in neurotransmitter release, leading to a greater abundance of functional synapses [35]. These actions contribute to essential brain functions such as learning, memory, and overall cognition [11,18,35,36,37]. Moreover, BDNF has been linked to participation in neuroinflammation through its ability to induce and be induced by NF-κB [14]. Additionally, BDNF exerts its neuroprotective effects by driving the expression of anti-apoptotic Bcl-2 family members and caspase inhibitors, causing inhibition of pro-apoptotic proteins like Bax and Bad, and enhancing the production of antioxidant enzymes while boosting the repair of damaged DNA in neurons [12].

## 2. Clinical Correlations

The alteration of BDNF levels, as well as the imbalance between pro-BDNF and m-BDNF, and deficits in BDNF signaling are associated with the pathogenesis of various neurologic and psychiatric disorders, including depressive disorder, schizophrenia, and bipolar disorder [8,11,23,38,39,40,41].

Research examining postmortem brain tissue from individuals diagnosed with bipolar disorder (BD) and major depressive disorder (MDD) revealed a decrease in BDNF levels in the hippocampus and prefrontal cortex, areas linked to memory processing [42]. However, the post-mortem brain tissue of individuals with schizophrenia showed dissimilar results, with some studies reporting BDNF increases in the prefrontal cortex and the hippocampus, while others noted BDNF decreases in these parts of the brain [42,43]. Additionally, an increase in the volume of the left hippocampus, as observed through magnetic resonance imaging (MRI), correlates positively with higher BDNF levels after eight months of atypical antipsychotic treatment in patients with first-episode schizophrenia [44]. Moreover, post-mortem brain tissue from patients who received antidepressant treatments demonstrated an increase in BDNF levels in the hippocampus and cortex [43,45].

Reduced BDNF blood levels have been reported in individuals with schizophrenia [43,46,47,48], bipolar disorder [43,48,49,50], and depression [51,52,53,54]. Moreover, patients who undergo antidepressant treatment tend to have increases in serum BDNF levels [52,53,54,55]. This effect has also been observed following electroconvulsive therapy (ECT) [51,56] and Repetitive Transcranial Magnetic Stimulation [51]. Importantly, patients with higher BDNF blood levels demonstrate a more positive response to antidepressant treatment, regardless of whether they had prior exposure to antidepressant medication [55]. Additionally, depressive episodes of greater than eight months’ duration correlated significantly with higher BDNF levels, which might indicate an attempted neuroprotective effect by specific brain structures in response to the perceived stress associated with depressive illness [55]. In contrast, schizophrenia patients exhibited a correlation between a greater decrease in BDNF blood levels and disease duration [57], whereas plasma BDNF, but not serum, levels were increased after antipsychotic treatment in schizophrenia patients [57]. Meta-analyses conducted by Fernandes et al. (2015b) showed that in bipolar disorder, both manic and depressive episodes are associated with decreased peripheral BDNF levels, while BDNF levels remain relatively stable during euthymic periods. They also noted a negative correlation between BDNF levels and the severity of manic and depressive symptoms. Moreover, their study revealed that peripheral BDNF levels increase in plasma, but not serum, following successful treatment of an acute manic episode, whereas treatment of a depressive episode did not show a similar effect [58].

Interestingly, higher BDNF levels are associated with later disease onset in adult BD patients, but not in pediatric patients. However, children who experienced the onset of BD before reaching puberty were found to have higher levels of BDNF compared to those who developed BD during puberty [59]. Additionally, a longer duration of BD was associated with higher peripheral blood BDNF concentrations [50], but no such correlation was found in a previous study [58]. Moreover, newly diagnosed patients with BD, as well as their unaffected first-degree relatives, had higher levels of BDNF compared to healthy individuals [60].

Lastly, the lower baseline levels of both plasma m-BDNF and the ratio of m-BDNF to pro-BDNF (M/P) in individuals with BD compared to those with MDD highlight the potential of the M/P ratio as a differential diagnostic biomarker for BD among patients experiencing depressive episodes [61].

## 3. Polymorphism

The combination and interplay of diverse BDNF and BDNF-AS variants might play a collaborative role in the development of psychiatric disorders [27]. BDNF gene variations can have a significant effect on brain structures and physiology, influencing memory and cognitive functions and leading to neurocognitive impairments and neuropsychiatric disorders [16,62,63]. For instance, rs6265 Met allele carriers have been shown to exhibit a smaller prefrontal cortex and hippocampus, which can contribute to decreased executive function and increased susceptibility to mood disorders [40,64]. Furthermore, the BDNF gene polymorphism may influence brain circuitry, potentially impacting individual responses to antipsychotic medication, which suggests the potential significance of BDNF genetic variants in predicting treatment outcomes [65].

Table 1 presents an overview of single nucleotide polymorphisms (SNPs) in human BDNF and BDNF-AS genes which we reviewed using the SNP database of the National Library of Medicine “https://www.ncbi.nlm.nih.gov/snp/ (accessed on 8 May 2023)”. We will provide a brief synopsis of the reported findings for each of these SNPs below.

### 3.1. SNP rs6265

The rs6265 at position chr11:27658369 (GRCh38.p14) exhibits the alleles C > T and is associated with a missense variant in the BDNF gene and a non-coding transcript variant in the BDNF-AS gene “https://www.ncbi.nlm.nih.gov/snp/rs6265 (accessed on 8 May 2023)”. Polymorphism rs6265, also known as G196A or Val66Met, is a single nucleotide substitution from guanine to adenine at position 196 of the BDNF gene. This substitution results in the conversion of valine (Val) to methionine (Met) at codon 66 in the N-terminal prodomain of the precursor BDNF protein (proBDNF). This variant disrupts BDNF transport, leading to a decrease in BDNF secretion and subsequently affecting its function, which contributes to neurocognitive impairments [45,66,67,68,69,70,71].

#### 3.1.1. SNP rs6265 and Depression

Based on a study involving individuals with MDD in Malaysia, Aldoghachi et al. (2019) discovered that possessing the BDNF rs6265 allele (A) increases the susceptibility to developing MDD, suggesting a potential contribution of BDNF to the origin of the disorder. The research also noted a significant decline in plasma BDNF levels among MDD subjects compared to controls. However, there was no indication of the influence of the rs6265 genotypes on the BDNF level [72]. In contrast, Caldieraro et al. (2018) found that the Met allele was associated with higher BDNF levels in individuals with MDD in a Brazilian population [73]. Interestingly, they also found that the Met allele was associated with lower levels of TNF-α [73], which is contradicted by a meta-analysis study that found MDD associations with higher levels of TNF-α in European individuals [73,74].

In another study conducted by Caldwell et al. (2013), involving male and female undergraduate students, it was shown that individuals with the Met allele and low levels of neglect exhibited higher depressive symptoms compared to those with the Val/Val allele. However, in cases of high neglect, depressive symptoms increased only in individuals with the Val/Val allele. Additionally, the study demonstrated that among Val/Val individuals, the relationship between neglect and depression was mediated by emotion-focused coping styles and reduced perceived control, while this mediation was not observed in Met carriers [75].

Shen et al. (2020) investigated the cortical thickness in MDD patients and found that individuals carrying the Met allele showed thinner cortical thickness in the left rostral anterior cingulate (rACC) compared to healthy controls carrying the same allele. Symptom severity and illness duration did not exhibit significant correlations with cortical thickness. These findings suggest that the presence of the Met allele may be a vulnerability factor for cortical thickness loss in the left rACC among MDD patients [76].

Val66Met BDNF polymorphism may affect white matter fiber tracts, which can be evaluated by fractional anisotropy (FA), assessing the direction and diffusion of water in brain tissue with the usage of MRI with diffusion tensor imaging (DTI). This technique helps to identify microstructural abnormalities and changes in white matter integrity [77]. According to Carballedo et al. (2012), individuals with MDD who possess the Met allele of the BDNF gene exhibit reduced FA in the uncinate fasciculus compared to MDD patients who are homozygous for the Val allele and healthy individuals carrying the Met allele. The study also found a significant three-way interaction involving the cingulum (dorsal, rostral, and parahippocampal regions), brain hemisphere, and BDNF genotype. Specifically, Met allele carriers demonstrated higher FA in the left rostral cingulum than Val/Val allele carriers. This suggests that the Met allele of the BDNF polymorphism may increase susceptibility to dysfunctions associated with the uncinate fasciculus, which is known to be involved in negative emotional-cognitive processing bias, autonoetic self-consciousness, and declarative memory issues. The findings highlight the significance of neurotrophic involvement in the connections between the limbic system and the prefrontal cortex [77].

In a prospective observational single-center cohort study conducted in China, researchers recruited early-stage breast cancer patients and found a significant link between the T/C and T/T genotypes of rs6265 and depressive symptoms in these patients [78]. Moreover, in a study by Soler et al. (2022), the authors examined how physical activity and stress influence depressive symptoms in individuals with various BDNF genotypes. Their findings revealed that individuals with the rs6265 AA genotype experienced a decrease in depressive symptoms as physical activity levels increased. However, those with the AA genotype were more susceptible to higher levels of depressive symptoms when exposed to elevated levels of childhood stress. Interestingly, AA carriers who encountered greater stress levels benefited from increased physical activity, which served as a protective factor against the presence of depressive symptoms [79].

According to a study conducted by Czira et al. (2012), the rs6265 polymorphism was linked to the intensity of depressive symptoms in an elderly population. Importantly, this connection was unrelated to neurotransmitter levels. Furthermore, the research revealed that this association specifically applies to older individuals aged 74 years or above, as well as men [80]. Additionally, Losenkov et al. (2020) conducted a study on newly admitted depressed patients who had not yet received antidepressant treatment. They discovered a notable correlation between the BDNF gene variant rs6265 and the intensity of depression. This finding suggests that the presence of the BDNF rs6265 variant may contribute to the severity of depression in this particular population [81].

In their meta-analysis, Pei et al. (2012) discovered a significant association between the Met allele of BDNF and geriatric depression. Based on their findings, they concluded that the BDNF Met allele serves as a risk factor for geriatric depression [82]. On the other hand, Sun et al. (2016) did not find any correlations between the rs6265 polymorphism and MDD in a Han Chinese population. However, they observed a significant interaction between carriers of the rs6265 (GG) genotype and the Serotonin Transporter (5-HTTLPR) heterozygous (LS) genotype, showing an increasing risk for MDD compared to other genotypes [83].

In a study by Herbert et al. (2012) involving premenopausal women at high risk of developing MDD, an interaction between cortisol and the Val66Met genotype was observed in relation to new depressive episodes. The researchers found a U-shaped relationship between the mean adjusted cortisol at baseline and depressive onsets specifically in Val/Val homozygotes, whereas this relationship was not evident in carriers of the Met allele. Higher cortisol values appeared to have a weak, non-significant protective effect against depression in Met allele carriers [84].

Examining the impact of the rs6265 polymorphism on individuals with MDD and healthy controls, Lisiecka et al. (2015) found that the allelic variations in this gene were associated with distinct neural activation patterns. Individuals with MDD who were homozygous for the Val allele exhibited decreased neural activation in areas responsible for cognitive appraisal of emotional scenes. On the other hand, individuals with MDD who carried the Met allele demonstrated increased activation in subcortical regions associated with visceral reactions to emotional stimuli. These findings suggest that the two allelic variants are linked to specific neural correlates of MDD, potentially indicating different mechanisms of the disorder in each group [85].

In a study conducted by Schosser et al. (2017) in patients with MDD who received at least four weeks of antidepressant treatment, a significant genotypic association was observed between the rs6265 polymorphism and an elevated risk of suicide in remitters. Additionally, a haplotypic association was found between the functional Val66Met polymorphism and rs10501087 in remitters who were at risk of suicide. These findings suggest that the rs6265 polymorphism may play a role in increasing the risk of suicide among patients with MDD [86].

According to a study conducted by Chen et al. (2021), patients suffering from treatment-resistant depression (TRD) who possessed the Val allele of the rs6265 polymorphism demonstrated a greater likelihood of responding positively to a low-dose ketamine infusion. The researchers found that the infusion of ketamine had rapid and long-lasting antidepressant effects on the affective and cognitive symptoms of depression, but not on the somatic symptoms [87].

In contrast, Musil et al. (2013), observed a potential but not statistically significant association between the rs6265 (Val66Met) polymorphism and overall treatment response in MDD patients. The response rates suggested that individuals with the GG (Val/Val) genotype had lower response rates compared to those with the GA (Val/Met) genotype, and the AA (Met/Met) genotype had the highest response rates [88]. Moreover, Liu et al. (2014) found a significant association between the rs6265 A allele and coronary artery disease-related depression, as well as higher responses to sertraline in A allele carriers [89]. Similarly, Ochi et al. (2019) found that premenopausal women (age < 50) with depressive disorder, who had not taken antidepressant medications for at least six months prior to the four-week study, showed a better response to antidepressant treatment during the last two weeks if they had the rs6265 AA/GA genotype compared to the rs6265 GG genotype [90]. Likewise, Pathak et al. (2022) discovered that the Val66Met polymorphism of the BDNF gene was associated with MDD, and patients carrying the Met allele exhibited a more favorable response to antidepressant treatment. Additionally, MDD patients had higher serum BDNF levels compared to healthy individuals, but lower serum BDNF levels in MDD patients were correlated with better results following ECT [91].

In a study by Schosser et al. (2022), it was discovered that the presence of the rs6265 polymorphism was significantly associated with treatment response in depressed patients undergoing cognitive behavioral therapy (CBT) within a standardized 6-week outpatient rehabilitation program [92]. Additionally, McClintock et al. (2020) found that the rs6265 polymorphism can moderate the neurocognitive effects of Transcranial Direct Current Stimulation (tDCS) treatment in a dose-dependent manner among participants with unipolar or bipolar depression. Specifically, patients with the Val/Val genotype showed greater improvement in verbal learning and memory under the high-dose condition, while those under the low-dose condition exhibited poorer results [93]. In contrast, Loo et al. (2018) did not find an association between the Val66Met polymorphism and treatment response to tDCS in unipolar and bipolar depression [94].

In a 12-week prospective longitudinal study, Li et al. (2013) identified significant gene-gene interactions involving the BDNF gene variant rs6265 and NTRK2 gene variants (rs1387923, rs2769605, and rs1565445) in Han Chinese patients with TRD. The results indicate that the interactions between the BDNF gene and the NTRK2 gene are likely to play a crucial role in the development of TRD [95].

According to Liang et al. (2018), their study revealed significant gene-gene interactions in patients with post-stroke depression (PSD) involving BDNF gene polymorphism (rs6265) in conjunction with p11 (rs11204922), tPA (rs8178895, rs2020918), and p75NTR (rs2072446, rs11466155). The findings suggest that the impact of BDNF on PSD may be influenced by these specific genetic polymorphisms within the intracellular signal transduction pathways [96].

In their study, Yang et al. (2016) explored the role of gene-environment interactions in the development of MDD. They discovered that a significant three-way interaction with the combination of the BDNF rs6265 G allele, PRKCG rs3745406 C allele, and a high level of negative life events was strongly associated with the occurrence of MDD [97].

#### 3.1.2. SNP rs6265 and Schizophrenia

Zakharyan et al. (2014) investigated the link between BDNF plasma levels and the rs6265 polymorphism in an Armenian population with schizophrenia. The study revealed that individuals affected by schizophrenia and carrying the minor allele of rs6265 (AG/AA) showed significantly lower BDNF levels compared to those who were homozygous for the standard G allele [98]. However, in contrast to these findings, Chen et al. (2014) did not observe any association between rs6265 variations and differences in plasma BDNF levels in participants with schizophrenia or BD from the Han Chinese population in Taiwan [99].

In their study, Zhang et al. (2018) observed no noteworthy differences in the frequency and distribution of rs6265 alleles and genotypes when comparing individuals with schizophrenia and those without. However, they did discover a significant association between the combination of the BDNF gene (rs6265 AA/AG) and the TNF-α gene (rs1799964 CC/CT) and the development of schizophrenia. Additionally, the research indicated significant differences in both attention and overall results on the Repeatable Battery for the Assessment of Neuropsychological Status (RBANS) among individuals with rs6265A and rs1799964C alleles and those who lacked them. These findings suggest that BDNF and TNF-α might contribute to an elevated vulnerability to schizophrenia and cognitive impairment [100].

The impact of rs6265 on the activation of the anterior cingulate cortex (ACC) and prefrontal regions (PFC), as well as the connectivity among these regions, was investigated by Schweiger et al. (2019) using functional Magnetic Resonance Imaging (fMRI). The results indicated a notable increase in connectivity between the ACC and PFC among individuals carrying the Met allele, especially in those with an increased familial susceptibility to schizophrenia. These findings suggest that this specific genetic variation may exert an influence on a neural circuit associated with cognitive control, potentially making individuals more vulnerable to the development of schizophrenia [101].

In the study conducted by Ping et al. (2022), no statistically significant differences were found in the genotype or allele frequencies of the rs6265 polymorphism between individuals diagnosed with schizophrenia and the control group. However, notable differences were observed in the Positive and Negative Symptom Scale (PANSS) concerning the general symptom G12, which pertains to judgment and lack of insight, among patients with various rs6265 genotypes of the BDNF gene. Additionally, the frequency of the haplotype rs6265(C)/rs11030101(A)/rs2030324(A) was significantly higher in individuals with schizophrenia compared to those in the control group [71].

According to the study of Suchanek et al. (2012, 2013), no significant correlation was found between the BDNF val66met polymorphism and the occurrence of schizophrenia. However, a connection was observed regarding the clinical course of the disease. Specifically, in males (but not females), the val/met genotype was associated with an earlier age at onset, while the val/val genotype was linked to more severe symptoms, particularly in the domain of general symptoms measured by the PANSS-G scale. Furthermore, the analysis of PANSS single-item scores indicated that individuals with the val/met genotype displayed increased intensity in hallucinatory behavior. Additionally, they discovered a significant linkage disequilibrium between rs6265 (val66met) and rs28383487 (C-281A), where the Met-C haplotype tended to lower incidence in schizophrenic patients than controls [46,102].

Li et al. (2013) reported that there were no statistically significant variations in allele and genotype distributions of the rs6265 polymorphism between schizophrenia patients and the control group in a Han Chinese population. However, by employing quantitative trait analysis, they did observe a significant correlation between the rs6265(A)/rs12273539(C)/rs10835210(A) haplotype and the severity of negative symptoms using the PANSS scale [103]. In contrast, Zhai et al. (2013) concluded that while the rs6265 polymorphism is not associated with susceptibility or cognitive function of schizophrenia in the Chinese population, it is significantly linked to clinical symptoms, particularly noting a higher frequency of the A/A genotype in patients with extrusive positive symptoms compared to those with negative symptoms [104].

Fu et al. (2020) reported a positive correlation between rs6265 and schizophrenia in Han Chinese patients from eastern China. Interestingly, they found that the minor allele (A) of rs6265 had a protective effect against schizophrenia. Additionally, the researchers identified several methylation sites near the BDNF gene that were linked to rs6265 [105].

Suchanek-Raif et al. (2020) conducted a case-control study to investigate the predisposition for schizophrenia. They discovered that a haplotype formed by five SNPs (rs1867283, rs10868235, rs1565445, rs1387923, and rs2769605) in the TrkB gene exhibited a protective effect in men. However, when the rs6265 SNP of the BDNF gene was added, a specific GTAGCG haplotype was observed, which demonstrated protective effects exclusively in women. These findings were particularly significant for the paranoid subtype of schizophrenia without concomitant depressive episodes [106].

According to Morozova et al. (2021), although no significant associations were found between BDNF rs6265 and schizophrenia in a case-control study, the presence of the Met66Met polymorphism was associated with higher scores in catalepsy, waxy flexibility, stupor, and negativism. Moreover, individuals with the TT genotype exhibited more pronounced impairments in frontal lobe function [107].

Zhang et al. (2017) examined the connection between cognitive impairment and the rs6265 polymorphism in individuals experiencing their first episode of schizophrenia. Their study revealed that patients diagnosed with schizophrenia exhibited notably lower scores in total IQ, verbal IQ, and performance IQ in comparison to the control group. The researchers identified genetic factors as contributors to the observed decrease in both total IQ and verbal IQ. Particularly noteworthy was the finding that patients carrying the Val/Val genotype demonstrated significantly lower verbal IQ scores when compared to Val/Met and Met/Met carriers. Furthermore, within the patient group, a negative correlation was observed between the total IQ of Val/Val carriers and the presence of positive symptoms and thought disorder. These results provide valuable insights into the relationship between cognitive impairment and the rs6265 polymorphism in individuals experiencing their first episode of schizophrenia [108].

Cheah et al. (2014) investigated the impact of BDNF polymorphism in both schizophrenia patients and alcohol-dependent individuals within Australian populations. Their study revealed significant allelic associations between the rs6265 polymorphism and comorbid alcohol dependence, as well as risk-taking behavior after consuming alcohol. Specifically, strong associations between rs6265 and alcohol dependence were observed in males with schizophrenia, while females exhibited associations with multiple behavioral measures indicative of repetitive alcohol consumption. Haplotype analysis further highlighted the association of the rs6265(A)/rs7103411(C) haplotype with comorbid alcohol dependence among patients diagnosed with schizophrenia [109].

In the Kim et al. (2018) study, no notable disparities were observed in the distribution of genotype or allele frequencies of rs6265 between individuals diagnosed with schizophrenia and the control group. Nonetheless, an elevated prevalence of suicide attempts was identified among patients with the A allele genotype, in comparison to those with the G/G genotype for rs6265. These findings suggest the potential utilization of the BDNF gene as a predictive indicator for attempting suicide in individuals with schizophrenia. Such information has the potential to aid in the management of schizophrenia patients [110].

In a study by Zai et al. (2012), the rs6265 (Val66Met) polymorphism was found to be associated with the response to antipsychotic treatment in schizophrenia patients of European descent. Specifically, the G/G genotype and G allele of Val66Met were more prevalent in the responder group compared to the non-responder group. Moreover, haplotypes involving Val66Met and rs1519480 were found to be linked to weight gain, with the G-A haplotype being significantly overrepresented in the group of individuals experiencing weight gain [65]. Moreover, Li et al. (2017) identified a significant correlation between the G allele of rs6265 and the alteration in waist-to-hip ratio induced by atypical antipsychotics in Chinese Han patients diagnosed with schizophrenia, who were monitored for a duration of 12 weeks [111]. In contrast, Bonaccorso et al. (2015) did not observe a significant difference in BMI between schizophrenia Met allele carriers and Val homozygous individuals [112].

The study by Zhang et al. (2013) revealed a significant correlation between the minor allele of rs6265 and the administration of clozapine therapy in patients diagnosed with schizophrenia. Individuals homozygous for the minor allele displayed a 4–8 times higher likelihood of being unresponsive to antipsychotic treatment compared to those homozygous for the major allele. Furthermore, it was observed that rs6265 exhibited a high degree of linkage disequilibrium with rs10501087 and rs11030104, leading to a robust association with resistance to antipsychotic treatment in carriers possessing the minor alleles. Notably, this elevated risk demonstrated a dose-dependent relationship [113].

#### 3.1.3. SNP rs6265 and Bipolar Disorder

According to the study by Park et al. (2015), children (aged 9–18) with the val/val genotype of biological parents diagnosed with BD displayed notably elevated levels of anxiety compared to BD offspring with val/met or met/met genotype. This finding suggests that subclinical anxiety symptoms in these children may serve as early prodromal indicators of BD and potentially early manifestations of an anxiety disorder [114].

Ivanova et al. (2013) found a tendency towards a higher occurrence of the C (Val) allele of rs6265 and borderline significance for the Met/Met genotype in patients with bipolar disorder, type I (BD-I), bipolar disorder type II (BD-II), and schizoaffective disorder, bipolar type. Additionally, the haplotype analysis revealed a higher frequency of the rs6265(C)-rs16917237(G) and rs6265(C)-rs16917237(G)-rs12273363(T) haplotypes in the patient group compared to healthy controls [115].

A study by Paul et al. (2021) revealed significant differences in the rs6265 polymorphism, notably observing a higher occurrence of the G (Val) allele among BD patients from India [116]. Additionally, in a recent study by Yoldi-Negrete et al. (2022), it was concluded that being an rs6265 Met carrier is a significant predictor of impaired functioning in individuals with BD-I [117]. In contrast, Wang et al. (2014a) found, no association between rs6265 and the risk of BD-I or treatment response. However, in a comprehensive meta-analysis conducted by Wang et al. (2014b), a significant association was found between the rs6265 Val allele and reduced vulnerability to BD-II [118].

Chang et al. (2013) conducted a study among Han Chinese in Taiwan and found that the rs6265 Val/Val genotype of the BDNF polymorphism correlated with BD-I comorbid with anxiety disorder (AD). They also observed an interaction between the rs6265 Val/Val genotype and the Gly/Gly genotype of the DRD3 Ser9Gly polymorphism in BD-II comorbid with AD, but not in BD-II without AD comorbidity, when compared to healthy controls. DRD3 is the dopamine D3 receptor gene on chromosome 3q13.3. These findings suggest that the rs6265 polymorphism is associated with BD-I comorbid with AD and may modulate the influence of the DRD3 Ser9Gly polymorphism in BD-II comorbid with AD [119]. Interestingly, Lee et al. (2013) discovered a significant interaction effect for the Val/Val genotypes of the BDNF Val66Met polymorphism and the Val/Met and Met/Met genotypes of the COMT Val158Met polymorphism among BD-II without AD patients compared to healthy controls [120].

Microribonucleic acids (miRNAs) have been shown to play important roles in many biological processes in the central nervous system, notably neurogenesis, synaptic plasticity, and neuroproliferation. In this context, the study by Wang et al. (2014a) revealed a significant gene-to-gene interaction between the BDNF rs6265 and MIR206 rs16882131 polymorphisms. This interaction confers susceptibility to BD-I and the response to mood stabilizers. Specifically, individuals with the rs6265 AA genotype and the miRNA-206 (MIR206) TT/TC genotype had a significantly lower average treatment score compared to those with the MIR206 (CC) genotype and rs6265 (AA, AG, GG) genotypes [121]. In addition, Lee et al. (2021) determined a significant association between miR221-5p and miR370-3p with BDNF levels in the rs6265 Val/Met genotype in patients with BD-II [122].

Nassan et al. (2015) identified a significant association between the rs6265 Met allele and early-onset BD in individuals aged 6–15 years. However, no substantial evidence of association was observed for the rs6265 SNP when comparing adult early-onset cases, specifically those with disease onset at or before 19 years of age, with control groups [123]. In a separate study on BD patients, Nassan et al. (2020) utilized pyrosequencing to analyze the methylation levels of two CpGs located in Promoter-IX, specifically focusing on CpG2, which includes the G(Val) allele of the Val66Met variance. The results revealed a significant increase in methylation at Promoter-IX/CPG-2, particularly in individuals with early onset BD when compared to the control group. Interestingly, this methylation difference was not observed in the late-onset cases when compared to the controls [124]. On the other hand, Schröter et al. (2020) conducted a study investigating the influence of the BDNF val66met genotype on BDNF DNA methylation changes and BDNF serum levels in acute and remitted phases of BD and MDD patients, compared to healthy controls. However, their study did not detect any influence of the genotype on either methylation or BDNF serum levels [125].

Lee et al. (2016) presented evidence suggesting a positive correlation between increases in plasma BDNF levels and improvements in the Wisconsin Card Sorting Test (WCST) scores among individuals with BD-I who carried Val/Met genotypes. However, it is important to note that this correlation did not maintain statistical significance after adjusting for multiple comparisons [126]. In contrast, Zeni et al. (2016a) found no significant association between the rs6265 polymorphism and BDNF serum levels in children and adolescents with BD comorbid with attention-deficit/hyperactivity disorder (ADHD) [127]. Moreover, Kenna et al. (2014) did not see any difference in val66met polymorphism and plasma BDNF concentrations between women with BD and healthy controls [128]. On the other hand, Grande et al. (2014) observed consistently lower serum BDNF levels in BD patients who were carriers of the rs6265 Met allele compared to Val homozygotes [129].

According to a study conducted by Pae et al. (2012), individuals diagnosed with BD who possessed the A allele and AA/AG genotypes of rs6265 exhibited a significantly younger age of onset for BD in comparison to carriers of the G allele and GG genotype. Additionally, the analysis of haplotypes indicated a potential link between various haplotypes and the age at which BD symptoms first appeared. Specifically, the combination of rs6265(A)/rs11030101(A) and rs6265(A)/rs11030101(A)/rs10835210(C) haplotypes was associated with a younger age of onset. On the other hand, the rs6265(G)/rs11030101(A) and rs6265(G)/rs11030101(A)/rs10835210(C) haplotypes were associated with an older age at onset [130].

In their study, Porcelli et al. (2017) found a significant association between rs6265 and the self-transcendence dimension of the Temperament and Character Inventory 125 items (TCI) in BD patients. Specifically, individuals carrying the TT genotype exhibited higher scores in self-transcendence compared to those with TC/CC genotypes [131]. In contrast, Rolstad et al. (2016) did not find any significant associations between rs6265 polymorphism and cognitive functioning in BD within the Swedish population [132].

Kennedy et al. (2022) reported a significant association between the rs6265 polymorphism and regional brain surface area in Caucasian youth (aged 13–20 years) diagnosed with BD. Through vertex-wise analysis, they found that individuals carrying the Met allele exhibited a larger surface area in the middle temporal gyrus and greater volume in the supramarginal gyrus compared to those with the Val/Val genotype. Moreover, the Met allele was specifically linked to increased surface area in the lateral occipital lobe among BD patients. Additionally, interaction effects were observed in the postcentral gyrus and supramarginal gyrus, with BD Met carriers showing a smaller surface area compared to Met carriers in the healthy control group [133].

In a systematic meta-analytical review by Harrisberger et al. (2015), it was found that the rs6265 SNP of the BDNF gene showed no significant association with hippocampal volumes in neuropsychiatric patients with MDD, anxiety, BD, or schizophrenia. Regardless of BDNF genotype, patients exhibited significantly smaller hippocampal volumes compared to healthy controls. The authors concluded that left, right, and bilateral hippocampal volumes were not significantly associated with rs6265 in neuropsychiatric patients. Interestingly, both Val/Val homozygotes and Met carriers from the patient group displayed smaller hippocampal volumes than healthy controls with the same allele, and these effects were consistent across genotypes [134].

Similarly, Zeni et al. (2016b) did not find any significant variations in the volume of the left or right hippocampus between individuals with BD and control subjects. However, they did discover a significant interaction between low scores on the family functioning cohesion subscale, as measured by the Family Environment Scale, and the presence of the rs6265 Met allele. This interaction was associated with left hippocampal volume in patients with BD, while no significant differences were observed in the right hippocampus [135].

Additionally, Cao et al. (2016) observed that individuals diagnosed with BD-I who carried the Met allele of the rs6265 variant exhibited smaller hippocampal volumes on MRI and demonstrated reduced memory performance on multiple California Verbal Learning Task (CVLT) scores compared to MDD patients and healthy controls [136]. Moreover, Pujol et al. (2020) found that the interaction between BDNF rs6265 and MTHFR rs1801133 SNPs polymorphism is a potential genetic risk factor for reduced hippocampal size, affecting both healthy controls and patients with first-episode psychosis, with rs6265 Met carriers and TT/CT genotype showing decreased hippocampal volume compared to CC genotype individuals [137].

Furthermore, Chang et al. (2018) conducted a study comparing BD-II patients with healthy individuals, and their findings indicate that BD-II patients who are Val homozygotes for the rs6265 gene scored significantly higher on the visual immediate memory subscale of the Wechsler Memory Scale (WMS) compared to the Met/Met and Val/Met individuals. On the other hand, the Met allele appears to play a specific role in memory dysfunction among BD-II patients, as Met homozygotes consistently scored lower than Val homozygotes on the WMS auditory delayed memory subscale, especially when compared to the healthy control group [138].

In their case-control study in a Mexican population, González-Castro et al. (2015) found no significant association between the Val66Met polymorphism and BD. However, when analyzing the relationship between rs6265 and a lifetime history of suicidal behavior in BD patients, they observed an association between the Val/Val genotype and suicide attempts [139].

Boscutti et al. (2022) observed no significant difference in impulsivity levels between BD patients and healthy controls when comparing rs6265 variations. Nevertheless, they discovered that individuals carrying the Met/Met genotype (but not Val/Met) in both groups exhibited decreased impulsivity levels, as assessed by the Barratt Impulsiveness Scale, in contrast to those with the Val/Val genotype [140].

Morales-Marín et al. (2016) found that in the Mexican population, BD patients with normal weight had a higher prevalence of the Met allele, while the Met/Met genotype was exclusively present in individuals with normal weight. These findings indicate a potential association between the presence of the Met allele and a decreased susceptibility to overweight and obesity among individuals diagnosed with BD [141]. In contrast, according to the study conducted by Bonaccorso et al. (2015), the rs6265 polymorphism was found to be a potential risk factor for weight gain, obesity, and lipid increases in patients with BD receiving antipsychotic treatment. Specifically, the researchers observed that individuals with BD who carried the Met allele, compared to those who were Val homozygous, exhibited the highest BMI at six months, along with an increase in the triglycerides/high-density (TRI/HDL) cholesterol ratio and log-triglycerides after three and six months of treatment with either risperidone or olanzapine [112].

Wang et al. (2012) conducted a study that identified a significant association between the rs6265 polymorphism and susceptibility to BD. The distribution of genotypes and alleles revealed a notable difference, with the Met allele identified as a risk factor for the disease. Additionally, the study evaluated the treatment response to lithium and valproate, finding contrasting effects of the Met allele on the response in BD-I and BD-II patients. In BD-I patients, the Val/Val genotype exhibited a significantly lower response score compared to Met allele carriers, while in BD-II patients, the Val/Val genotype showed a significantly higher response score compared to Met allele carriers [142]. However, in their meta-analysis, Ehret et al. (2013) concluded that the Val66Met polymorphism cannot predict lithium treatment response in BD patients [143]. Similarly, Paul et al. (2021) did not find a significant association between the rs6265 polymorphism and the response to lithium treatment in BD patients from India [116]. Interestingly, Lee et al. (2014) found a significant association between the addition of memantine to valproic acid in Val66Met heterozygote carriers treated for BD-II, resulting in a decrease in depressive symptoms compared to the use of valproic acid with a placebo [144].

### 3.2. SNP rs11030101

The rs11030101 at position chr11:27659197 (GRCh38.p14) exhibits the alleles A > G/A > T and is associated with an intron variant in the BDNF gene and a non-coding transcript variant in the BDNF-AS gene “https://www.ncbi.nlm.nih.gov/snp/rs11030101 (accessed on 8 May 2023)”.

Illi et al. (2013) conducted a study in Finnish patients diagnosed with MDD and found no significant differences in rs11030101 variations between the patient group and the control group. The study also did not identify any significant associations between this SNP polymorphism and remission or treatment response to selective serotonin reuptake inhibitors in their patient population [145]. However, Viikki et al. (2013) observed that individuals with the TA genotype of rs11030101, suffering from TRD, had diminished response to ECT compared to those with the TT genotype. This study suggests a potential association between ECT efficacy and BDNF polymorphism [146].

Pae et al. (2012) reported a potential association between the rs11030101 AT genotype and BD as well as schizophrenia. Their findings indicate a higher frequency of the rs11030101 AT genotype in individuals with these disorders compared to healthy controls and patients diagnosed with MDD [130]. Still, in a recent study by Ping et al. (2022), no significant variance in genotype or allele frequencies for rs11030101 was observed between schizophrenic patients and controls. Nonetheless, they did find differences in negative scale scores among schizophrenic patients with variations in the rs11030101 genotype. Furthermore, PANSS negative symptom scores were affected by different genotypes, with the AA genotype having the most pronounced impact and the TT genotype showing the least effect. Importantly, schizophrenic patients with the AA genotype at rs11030101 exhibited more prominent clinical negative symptoms [71].

### 3.3. SNP rs11030104

The rs11030104 at position chr11:27662970 (GRCh38.p14) exhibits the alleles A > G and is associated with intron variants in both the BDNF and BDNF-AS genes “https://www.ncbi.nlm.nih.gov/snp/rs11030104 (accessed on 8 May 2023)”.

In the study conducted by Calabrò et al. (2018) involving patients of Korean ancestry, no significant correlation was found between alterations in rs11030104 and the outcomes of response, remission, or resistance to antidepressant therapy among individuals diagnosed with MDD. Similarly, no substantial association was identified between variations in rs11030104 and response or remission achieved through treatment with mood stabilizers in patients with BD. However, intriguing patterns were noted, suggesting a possible link between the rs11030104 polymorphism and the reaction to antidepressant therapy in individuals diagnosed with MDD, as well as a correlation with the risk of developing BD [147]. Additionally, In the study by Schosser et al. (2022), the rs11030104 polymorphism exhibited a significant association with treatment response to cognitive behavioral therapy in depressed patients. Notably, this association remained significant even after utilizing the false discovery rate method for multiple testing corrections [92].

Zai et al. (2012) found a noteworthy relationship between the rs11030104 polymorphism and response to antipsychotic treatment in patients of European ancestry diagnosed with schizophrenia. Their study revealed a higher occurrence of the rs11030104 T/T genotype and T allele among responders compared to non-responders [65]. Moreover, Zhang et al. (2013) observed an increased risk of antipsychotic treatment resistance in schizophrenia patients on clozapine therapy carrying the minor allele for rs11030104. There was also a high linkage disequilibrium of rs11030104 with rs10501087 and rs6265, which could influence these findings [113]. Furthermore, Li et al. (2017) found a significant association between the rs11030104 G allele and the change of waist-to-hip ratio (WHR) induced by atypical antipsychotics in schizophrenic patients of the Chinese Han population who were followed for 12 weeks [111].

### 3.4. SNP rs10835210

The rs10835210 at position chr11:27674363 (GRCh38.p14) exhibits the alleles C > A/C > G and is associated with intron variants in both the BDNF and BDNF-AS genes “https://www.ncbi.nlm.nih.gov/snp/rs10835210 (accessed on 8 May 2023)”.

Significant differences in rs10835210 genotype frequencies were observed by Pae et al. (2012) among different patient groups of Korean ethnicity, demonstrating a greater prevalence of the rs10835210 (CA) genotype in individuals with BD and schizophrenia in comparison to MDD patients and healthy controls [130]. In contrast, Li et al. (2013) did not find any significant associations in the allele and genotype distributions of rs10835210 between schizophrenic patients and the control group in a Han Chinese population. However, they suggested that the rs10835210(A)/rs6265(A)/rs12273539(C) haplotype could potentially facilitate negative symptoms in schizophrenic patients [103].

On the other hand, Zhang et al. (2016a, 2016b) later conducted a study in the Han Chinese population and identified significant genotype and allele frequency differences for rs10835210 between schizophrenia patients and healthy controls. Haplotype analysis revealed higher frequencies of haplotypes containing the mutant A allele in schizophrenia patients compared to controls. Furthermore, patients carrying the mutational allele A exhibited more positive symptoms and a significant impact on language performance [148,149].

### 3.5. SNP rs2049046

The rs2049046 at position chr11:27702228 (GRCh38.p14) exhibits the alleles T > A and is associated with an intron variant in the BDNF gene “https://www.ncbi.nlm.nih.gov/snp/rs2049046 (accessed on 8 May 2023)”.

In a study of patients undergoing treatment for acute depressive episodes, Hennings et al. (2019) discovered that the rs2049046 variant had a significant impact on antidepressant response. The T allele was found to be beneficial for the antidepressant treatment results. They also observed lower cortisol responses to the combined dexamethasone suppression/corticotropin-releasing hormone challenge (dex/CRH) test at discharge for T allele carriers of rs2049046. In contrast, non-carriers of the T allele had the highest cortisol levels. These findings suggest an interaction between the HPA axis and the BDNF gene in MDD [27].

Liang et al. (2018) genotyped sixty-five single nucleotide polymorphisms (SNPs) in six genes (p11, tPA, PAI-1, BDNF, TrkB, and p75NTR) of the p11/tPA/BDNF pathway with minor allele frequencies > 5% from an initial series of 76 SNPs. They found significant gene-gene associations between BDNF SNP rs2049046 and p11 (rs11204922 SNP), tPA (rs8178895, rs2020918 SNPs), p75NTR (rs2072446, rs11466155), and TrkB (rs7816 SNP) genes. This gene-gene correlation was associated with an increased occurrence of post-stroke depression in Chinese patients with acute ischemic stroke [96].

### 3.6. SNP rs61888800

The rs61888800 at position chr11:27700731 (GRCh38.p14) exhibits the alleles G > C/G > T and is associated with an intron variant in the BDNF gene “https://www.ncbi.nlm.nih.gov/snp/rs61888800 (accessed on 8 May 2023)”.

Andre et al. (2018) conducted a study in Finnish patients with MDD who received selective serotonin reuptake inhibitor treatment for a period of 6 weeks and were assessed using the 107-item Temperament and Character Inventory temperament questionnaire (version IX) as well as the Montgomery–Åsberg Depression Rating Scale (MADRS). They found that the T-carriers of BDNF rs61888800 exhibited a more positive shift in temperament on the novelty-seeking (NS) dimension during the recovery phase compared to GG carriers. All patients included in the study had an MDD episode of at least moderate severity, and the correlation between rs61888800 T-carrying status and change in NS score from baseline to six weeks was significant. The authors suggested that this result could be due to variations in the expression of this SNP polymorphism in the brain. T-carriers of rs61888800 demonstrated lower expression of BDNF in the frontal cortex and higher expression in the cerebellum, with an increasing trend for the T-allele expression in the putamen [150].

In contrast to this, a previous study by Illi et al. (2013) did not find any associations between rs6188880 and MDD in Finnish-origin patients treated with selective serotonin reuptake inhibitors. Moreover, the study by Viikki et al. (2013) also did not observe any associations between rs61888800 and the efficacy of ECT in patients with MDD [146].

### 3.7. SNP rs2030324

The rs2030324 at position chr11:27705368 (GRCh38.p14) exhibits the alleles A > G and is associated with an intron variant in the BDNF gene “https://www.ncbi.nlm.nih.gov/snp/rs2030324 (accessed on 8 May 2023)”.

Pae et al. (2012) found no significant differences in genotype and allele frequencies of rs2030324 among Korean patients with MDD, BD, schizophrenia, and healthy controls [130]. However, Yang et al. (2013) discovered a significant association between rs2030324 in BDNF and schizophrenia in Han Chinese individuals from Southern China. Specifically, the rs2030324 (CC/CT) genotype increased the risk of schizophrenia. Additionally, rs2030324 in BDNF exhibited a significant combined effect with rs520688 in Neurogenic locus notch homolog protein 4 (NOTCH4). The rs2030324 (CC/CT) and rs520688 (AA/GG) variants were more common in schizophrenia cases, indicating them as risk markers for the disorder [151].

Nevertheless, Su et al. (2021) reported no significant differences in allele and genotype frequencies of rs2030324 among drug-naïve first episode (DNFE) patients, chronic patients with schizophrenia, and healthy controls in China. However, they did find that DNFE patients with TT and TC genotypes performed worse in language, attention, and total scores on the Repeatable Battery for the Assessment of Neuropsychological Status compared to chronic patients and healthy controls. After applying the Bonferroni correction, only the interaction effect on language remained significant. DNFE patients with rs2030324 (TT/TC) genotypes exhibited poorer language performance than chronic patients, but there was no significant difference in rs2030324 (CC) genotypes between DNFE and chronic patients. These findings suggest an important role of rs2030324 variants in the pathophysiology of the early stages of schizophrenia [152].

By contrast, a recent study by Ping et al. (2022) did not identify any significant differences in genotype or allele frequencies of rs2030324 between Han Chinese schizophrenic patients and control subjects. Nevertheless, they did observe a significant increase in the haplotype frequency of rs2030324(A) with rs11030101(A) and rs6265(C) in schizophrenic patients compared to controls [71].

### 3.8. SNP rs10501087

The rs10501087 at position chr11:27648561 (GRCh38.p14) exhibits the alleles T > C/T > G and is associated with an intron variant in the BDNF-AS gene “https://www.ncbi.nlm.nih.gov/snp/rs10501087 (accessed on 8 May 2023)”.

Zhang et al. (2013) noted that carrying the minor allele of rs10501087 increased the risk of antipsychotic treatment resistance in Caucasian schizophrenia patients. The minor allele homozygotes, in contrast to the major allele homozygotes, were more likely to be on clozapine therapy. Moreover, rs10501087 showed high linkage disequilibrium with rs11030104 and rs6265 [113].

Schosser et al. (2017) found that the C allele of rs10501087 showed a significant genotypic association with suicide risk in MDD patients who achieved remission after four weeks of treatment with antidepressants at an adequate dose. Additionally, they found a significant haplotypic association with rs6265 in the same group of patients [86].

In a later study, Schosser et al. (2022) further demonstrated that the rs10501087 polymorphism remained significantly associated with treatment response to cognitive behavioral therapy in depressed patients, even after multiple testing corrections applying the method of false discovery rate [92].

### 3.9. SNP rs12273539

The rs12273539 at position chr11:27661764 (GRCh38.p14) exhibits the alleles C > T and is associated with intron variants in both the BDNF and BDNF-AS genes “https://www.ncbi.nlm.nih.gov/snp/rs12273539 (accessed on 8 May 2023)”.

Yang et al. (2013) found no significant difference in the rs12273539 polymorphism between schizophrenia patients and healthy controls among Han Chinese in Southern China [151]. Similarly, Li et al. (2013) also did not find any significance, although they observed a significant association in the haplotype frequencies of rs12273539(C)/rs10835210(A)/rs6265(A) in schizophrenic patients with negative symptoms compared to healthy controls [103]. However, subsequent research by Zhang et al. (2016b) demonstrated an association between rs12273539 and cognitive performance in both schizophrenia patients and healthy controls in the Han Chinese population, showing worse results on attention index and RBANS total scores in schizophrenia patients homozygous for the T/T mutation [149,152].

In a recent study, Su et al. (2021) found no significant difference in genotype and allele frequencies of rs12273539 between schizophrenia patients and healthy controls, but a weak association of rs12273539 variations with neurocognitive functions was observed in both schizophrenia patients and healthy controls [152].

### 3.10. SNP rs7124442

The rs7124442 at position chr11:27655494 (GRCh38.p14) exhibits the alleles C > G/C > T and is associated with 3 Prime UTR Variant in the BDNF gene and an intron variant in the BDNF-AS gene “https://www.ncbi.nlm.nih.gov/snp/rs7124442 (accessed on 8 May 2023)”.

Calabrò et al. (2018) found trends of rs7124442 polymorphism association with response to mood stabilizers in BD patients [147]. Moreover, Ochi et al. (2019) demonstrated an association of rs7124442 with antidepressant drug response in depressive disorder patients who had not been treated with antidepressant medications at least six months before starting to receive it during the four-week study. rs7124442 (CC) post-menopausal (≥50 years) depressed patients responded significantly worse in the last two weeks of treatment compared to rs712442 (TT/CT), but not pre-menopausal women (<50 years of age) with depression [90]. However, other studies did not find any significant associations between rs7124442 variations and depression [80,81,88,147].

### 3.11. SNP rs1519480

The rs1519480 at position chr11:27654165 (GRCh38.p14) exhibits the alleles C > A/C > G/C > T and is associated with an intron variant in the BDNF-AS gene “https://www.ncbi.nlm.nih.gov/snp/rs1519480 (accessed on 8 May 2023)”.

Zai et al. (2012) observed an association between the rs1519480 polymorphism and weight change after using an antipsychotic medication in schizophrenia patients of European ancestry. Specifically, the A allele of rs1519480 and the A/A genotype were over-represented in schizophrenia patients who experienced weight gain. Additionally, the same association of antipsychotic-induced weight gain was seen in the rs1519480(A)-rs6265(G) haplotype [65]. Moreover, in BD patients of Korean ancestry, Calabrò et al. (2018) found trends of the rs1519480 polymorphism being associated with response to mood stabilizers in single SNP and haplotype analyses with rs7124442 [147].

Salehi et al. (2013) investigated the correlation between rs1519480 and N-acetyl-aspartate (NAA) levels in the brain. They identified a significant interaction between rs1519480 and age in relation to NAA levels, with the effect reaching significance at or above the age of 34.17. They observed a decline in NAA levels with advancing age among individuals with the TT genotype, but not among those with the CT or CC genotypes. Additionally, the study revealed a link between the T allele of rs1519480 and reduced expression of BDNF mRNA in the prefrontal cortex. These findings indicate that the rs1519480 SNP influences BDNF expression, thereby impacting prefrontal NAA levels over time. This genetic mechanism may contribute to dissimilarities in cognitive performance and susceptibility to various psychiatric disorders [153].

NAA performs a distinct function in facilitating lipid synthesis in myelin sheaths, enabling the transfer of acetate groups from neurons to oligodendrocytes [154]. Its crucial role in myelin synthesis underscores its significance, as reduced NAA levels can potentially impede the synthesis or restoration of myelin, thereby impacting white matter [155]. Interestingly, a growing body of research has focused on the association between NAA levels and various psychiatric disorders. Notably, diminished NAA levels are frequently observed in both individuals with schizophrenia and their unaffected first-degree relatives [155].

Proton magnetic resonance spectroscopy (1H-MRS) studies in people with chronic schizophrenia revealed that NAA levels were significantly lower in the frontal lobe, hippocampus, temporal lobe, thalamus, and parietal lobe compared to healthy people. Similarly, in first-episode psychosis, NAA levels were lower in the frontal lobe, anterior cingulate cortex, and thalamus, while lower NAA levels were observed in the hippocampus among individuals at high risk for psychosis [156].

In individuals with MDD, 1H-MRS detected decreased NAA levels in the right frontal and right parietal lobes, with a comparatively lesser extent of reduction in the left frontal lobe [157]. In young patients (14–22 years) with TRD, 1H-MRS showed lower concentrations of NAA in the left hippocampus, as well as a smaller left hippocampal volume [158].

In both depressive and euthymic periods of BD patients, 1H-MRS revealed a significant reduction of NAA levels in the left white matter prefrontal cortex. Additionally, during depressive periods, there was a significant increase in NAA levels observed in the left dorsolateral prefrontal cortex. The decline in NAA within the white matter of the prefrontal cortex is in line with the identified changes in neuroplasticity and synaptic plasticity seen in BD patients [154].

### 3.12. SNP rs11030094

The rs11030094 at position chr11:27638228 (GRCh38.p14) exhibits the alleles G > A and is associated with an intron variant in the BDNF-AS gene “https://www.ncbi.nlm.nih.gov/snp/rs11030094 (accessed on 8 May 2023)”.

In their study of patients admitted to the clinic for the treatment of acute depressive episodes, Hennings et al. (2019) discovered a significant effect of rs11030094 on antidepressant response, with the G allele considered beneficial for antidepressant response. Furthermore, they found significantly lower cortisol responses to the combined dexamethasone suppression/corticotropin-releasing hormone challenge test at discharge for G allele carriers of rs11030094. The interaction analysis of the SNP revealed the highest cortisol levels in subjects who were non-carriers of the G allele. These findings provide evidence that rs11030094 impacts HPA axis regulation in MDD. Notably, this genetic effect on the HPA axis was observed under antidepressant treatment, independent of medication and severity of depressive symptoms [27].

### 3.13. SNP rs28383487

The rs28383487 at position chr11:27722009 (GRCh38.p14) exhibits the alleles G > T and is associated with a 5 prime UTR variant in the BDNF gene and 2KB Upstream variant of uncharacterized LOC124902652 (ncRNA) gene “https://www.ncbi.nlm.nih.gov/snp/rs28383487 (accessed on 8 May 2023)”.

The study conducted by Suchanek et al. (2012) identified a correlation between the rs28383487 (C-281A) polymorphism and the timing of the initial episode of paranoid schizophrenia within the Polish population. In men, the presence of the rs28383487 (CA) variant was linked to a delayed onset of paranoid schizophrenia, while no such correlation was observed in women. Moreover, the study revealed a strong linkage disequilibrium between rs28383487 and val66met polymorphisms. Notably, the C-Met haplotype demonstrated a trend towards significantly lower frequency in the group of individuals with schizophrenia compared to the control group [46].

### 3.14. SNP rs16917237

The rs16917237 at position chr11:27680836 (GRCh38.p14) exhibits the alleles G > T and is associated with intron variants in both the BDNF and BDNF-AS genes “https://www.ncbi.nlm.nih.gov/snp/rs16917237 (accessed on 8 May 2023)”.

According to a study conducted by Ivanova et al. (2013), significant differences were observed in the G allele of rs16917237, which was found to be more prevalent among patients with BD-I, BD-II, and schizoaffective disorder bipolar type. Conversely, the T/T genotype was found to be less frequently represented in these patient groups. Furthermore, the analysis of haplotypes revealed that the haplotypes rs16917237(G)-rs6265(C), rs16917237(G)-rs12273363(T), and rs16917237(G)-rs6265(C)-rs12273363(T) were more commonly observed in the patient groups compared to the healthy control group [115].

## 4. Discussion

BDNF/BDNF-AS gene polymorphisms are very important factors to consider, as they can alter BDNF function causing multiple human phenotypes. For instance, BDNF polymorphisms of five SNPs (rs11030101, rs10835210, rs2049046, rs2030324, and rs2883187) showed an association with psychomotor speed, which is related to velocity of nerve conduction, data retrieval from memory, and speed of information processing, influencing the efficacy of most cognitive domains, with the best results observed in high-frequency allelic homozygotes [159].

Variations in the BDNF gene can negatively alter BDNF gene expression and function, leading to multiple consequences. These variations can result in improper folding of the BDNF protein, disrupt protein-protein interactions, affect binding affinities, impact protein localization, compromise the conformational stability of the protein, and reduce the ability of mature BDNF to bind to its TrkB receptor [160]. One specific example is the rs6265 polymorphism, which involves a change from valine to methionine. This alteration affects the packaging of BDNF in secretory granules, leading to a decrease in BDNF release-dependent activity. Furthermore, this polymorphism has been linked to altered stress responses, cognition, episodic memory, and a reduction in hippocampal activity ultimately impacting mood and facilitating the development of psychiatric consequences [161,162,163]. For instance, healthy carriers of the rs6265 Met allele showed a lower average verbal comprehensive index, a lower average performance reasoning index, and a lower average full-scale IQ-4, and additionally, an increased body mass index, decreased systolic and diastolic blood pressure [164].

Interestingly, researchers have found that individuals with the rs6265 Met/Met genotype demonstrated lower verbal IQ, while those with Val/Val and Val/Met genotypes showed average values for these measures [164]. However, it is important to note that there seems to be a contradiction regarding the association between BDNF polymorphism and intelligence. One study showed different results, indicating that first-episode schizophrenia patients with the Val/Val genotype had significantly lower verbal IQ scores compared to Val/Met and Met/Met carriers [108]. Additionally, in a separate study increased performance of Met carriers was observed on the vocabulary, block design, and object assembly subtests of the Wechsler Intelligence scale in comparison to the Val/Val genotype, regardless of diagnosis [165]. There are other studies that did not find any association between rs6265 polymorphism with cognitive function in schizophrenia [104] and BD [132].

Environmental changes have a strong impact on the BDNF gene, potentially influencing the study results [34]. It is important to consider the influence of potential epigenetic factors on the BDNF gene and cognition. For instance, in patients with psychoses, possessing the rs6265 Met allele along with high levels of childhood abuse was found to be significantly associated with poorer cognitive functioning compared to homozygotic Val carriers. Among Met carriers, there were noteworthy negative correlations between physical abuse and verbal abilities, as well as working memory, and between emotional abuse and executive function and verbal fluency. Additionally, in Met carriers who experienced childhood sexual abuse, there was a decrease in right hippocampal volume and an increase in right and left lateral ventricles [166].

These findings underscore the importance of considering gene-environment interactions in understanding the impact of BDNF SNPs on the development of mental health disorders. Indeed, there are other studies that reveal a complex interplay of gene-environment interactions, where the outcomes associated with specific alleles depend on the presence or absence of certain environmental influences. Moreover, the impact of alleles is not uniform across all environmental conditions. For instance, Met allele carriers with low levels of neglect display higher depressive symptoms compared to those with the Val/Val genotype. However, in cases of higher neglect, depressive symptoms increase only in individuals with the Val/Val genotype [75]. Furthermore, individuals homozygous for the Met allele show an increased predisposition to higher levels of depressive symptoms when exposed to elevated levels of childhood stress. However, interestingly, the same Met/Met genotype experiences a decrease in depressive symptoms with an increase in physical activity [79].

Indeed, physical activity has demonstrated a significant influence on BDNF. Szuhany et al. (2015) conducted a meta-analytic review, revealing that exercise can influence BDNF activity. Each exercise session contributes to a moderate increase in BDNF levels, and consistent physical activity can further amplify this effect. While regular exercise affects resting BDNF levels, its impact appears less pronounced compared to the immediate post-exercise effect and may not be considered robust. Notably, females demonstrated a less pronounced increase in BDNF levels following exercise compared to males [167]. However, another study found no increase in resting serum BDNF levels after six months of aerobic training, resistance training, or their combination in adolescents with overweight and obesity [168]. In contrast, healthy males aged 53 to 64 years demonstrated a reduction in BDNF serum levels following 12 weeks of aerobic and resistance training [169]. On the other hand, healthy older individuals experienced a significant immediate increase in serum BDNF levels following a 35-min physical exercise session [170].

Moreover, participation in regular cardiovascular and coordination training was found to be associated with hippocampal total volume increase, demonstrated on MRI scans, after the 12-month intervention period in 62–79 year-old adults [171]. However, another study on cognitively healthy individuals above the age of 60 revealed that people with the Val/Val genotype and higher levels of physical activity exhibited greater hippocampus and temporal lobe volumes on MRI. In contrast, Met allele carriers with increased physical activity demonstrated smaller temporal lobe volume [172]. Furthermore, individuals homozygous for the Val allele above the age of 55 demonstrated improved episodic memory performance as a result of physical activity, when compared to both inactive Val homozygous individuals and active Met carriers [173]. These findings underscore the significance of personalized medicine in the realm of healthcare. Understanding how exercise and genetics impact BDNF levels and the brain in different people can help to develop personalized treatment plans rather than applying a one-size-fits-all approach.

Additionally, it is worth noting that diet and nutrition influence BDNF expression and the brain. For instance, lower hippocampal BDNF expression and decreased spatial working memory were seen in the sedentary group of rats on a high-fat, high-sucrose Western diet in comparison to the sedentary group of rats on a plant-based, amylose/amylopectin blend, lower-fat diet [174]. Interestingly, higher consumption of a Western-style dietary pattern consisting of higher saturated fats and refined carbohydrates was found in correlation with smaller left hippocampal volume on MRI scans in individuals aged above 60 years, in comparison to individuals on healthy dietary patterns [175].

Beyond the interplay between BDNF alleles and the environment, gene-gene interactions emerge as another crucial factor influencing study results and predisposition to mental health disorders. Likewise, a three-way interaction was described, showing the combined effect of the BDNF rs6265 G allele, a high level of negative life events, and the PRKCG rs3745406 C allele, which exhibited a strong association with the development of MDD [97].

Furthermore, there are multiple SNPs within the BDNF gene itself and the BDNF-AS gene that exhibit strong linkage disequilibrium and interact with each other, potentially influencing susceptibility to psychiatric conditions and treatment response. For instance, in a study by Ping et al. (2022), no significant difference in the polymorphisms of individual SNPs, such as rs6265, rs11030101, and rs2030324, was observed between schizophrenic patients and controls. However, the frequency of a haplotype composed of rs6265(C)/rs11030101(A)/rs2030324(A) was significantly increased in schizophrenic patients compared to controls [71]. Furthermore, Zhang et al. (2013) reported an increased risk of antipsychotic treatment resistance in schizophrenia patients who are minor allele carriers of rs6265, rs11030104, and rs10501087. These SNPs demonstrated high linkage disequilibrium with each other, which could potentially influence the results of the findings [113].

Another factor that can potentially impact research on BDNF gene polymorphism is the variation in the genetic makeup of the studied population. For instance, the rs6265 Met allele is more prevalent in Asian populations compared to Caucasians, whereas it is nearly absent in Native American and African populations [7]. The persistence of the Met allele throughout evolution, along with its higher occurrence in populations outside Africa, indicates that it may have multiple functions, including both advantageous and disadvantageous aspects [176]. This variation may contribute to inconsistencies in study results. For example, Wang et al. (2023) suggested that the BDNF Val66Met gene polymorphism contributes to major depressive disorder in Caucasian populations, but this association was not observed in Asian populations. The conflicting outcomes could be attributed to variations in the frequency of the same allele among populations, the diverse effects of the allele on different races and age groups, and the influence of distinct environmental factors [177].

Among neuropsychiatric diseases, rs6265 is the most extensively studied BDNF SNP [178], and its polymorphism has shown associations with diverse mental health conditions. It is crucial to explore BDNF variants beyond the widely investigated Val66Met polymorphism [27]. Nevertheless, multiple other BDNF/BDNF-AS SNPs remain unexplored regarding their potential association with psychiatric disorders. For this review article, we selected 14 SNPs that have demonstrated significant findings in relation to depression, schizophrenia, or bipolar disorder. It is important to note that the scope of our review is limited to studies published from 2012 and identified through PubMed citations.

There are also SNPs that are only associated with the BDNF-AS gene, such as rs10501087, rs1519480, and rs11030094, which could be interesting to study more deeply to determine if their polymorphisms have independent effects from BDNF gene SNPs. For example, as mentioned earlier, rs10501087 showed high linkage disequilibrium with rs6265 and rs11030104 SNPs, which are linked to both the BDNF and BDNF-AS genes. Moreover, the T allele of rs1519480 was reported to be associated with lower BDNF mRNA expression and low NAA levels in the PFC of individuals in a postmortem study [153]. Additionally, rs11030094 polymorphism showed an impact on HPA axis regulation in MDD, and the rs11030094 G allele was reported to be beneficial for antidepressant response [27].

The research findings reviewed in this article underscore the intricate interplay between SNPs of BDNF and BDNF-AS genetic variations, gene-gene interactions, environmental factors, and their relation to mental health disorders and treatment outcomes in diverse patient populations.

In conclusion, numerous studies have established significant connections between specific variations of BDNF/BDNF-AS gene polymorphisms and psychiatric disorders such as bipolar disorder, schizophrenia, and major depressive disorder. These polymorphisms are linked to varied clinical characteristics, treatment responses, and outcomes. While some inconsistencies exist across studies, these discrepancies mainly arise from the natural diversity among populations and potential confounding factors, highlighting the dynamic nature of genetic influences on mental health disorders. Further research is needed to validate BDNF/BDNF-AS gene polymorphisms as biomarkers for psychiatric disorders, as it can potentially improve diagnostic accuracy and enable the personalization of therapeutic interventions based on individual genetic profiles in mental health disorders.

## Figures and Tables

**Table 1 jpm-13-01395-t001:** BDNF and BDNF-AS SNPs.

SNP	Gene: Consequence	Genomic Position	Alleles
rs6265	BDNF: Missense Variant	chr11:27658369 (GRCh38.p14)	C > T
	BDNF-AS: Non-Coding Transcript Variant		
rs11030101	BDNF: Intron Variant	chr11:27659197 (GRCh38.p14)	A > G/A > T
	BDNF-AS: Non-Coding Transcript Variant		
rs11030104	BDNF: Intron Variant	chr11:27662970 (GRCh38.p14)	A > G
	BDNF-AS: Intron Variant		
rs7124442	BDNF: 3 Prime UTR Variant	chr11:27655494 (GRCh38.p14)	C > G/C > T
	BDNF-AS: Intron Variant		
rs10835210	BDNF: Intron Variant	chr11:27674363 (GRCh38.p14)	C > A/C > G
	BDNF-AS: Intron Variant		
rs12273539	BDNF: Intron Variant	chr11:27661764 (GRCh38.p14)	C > T
	BDNF-AS: Intron Variant		
rs16917237	BDNF: Intron Variant	chr11:27680836 (GRCh38.p14)	G > T
	BDNF-AS: Intron Variant		
rs2049046	BDNF: Intron Variant	chr11:27702228 (GRCh38.p14)	T > A
rs61888800	BDNF: Intron Variant	chr11:27700731 (GRCh38.p14)	G > C/G > T
rs2030324	BDNF: Intron Variant	chr11:27705368 (GRCh38.p14)	A > G
rs28383487	BDNF: 5 Prime UTR Variant	chr11:27722009 (GRCh38.p14)	G > T
	LOC124902652: 2KB Upstream Variant		
rs10501087	BDNF-AS: Intron Variant	chr11:27648561 (GRCh38.p14)	T > C/T > G
rs11030094	BDNF-AS: Intron Variant	chr11:27638228 (GRCh38.p14)	G > A
rs1519480	BDNF-AS: Intron Variant	chr11:27654165 (GRCh38.p14)	C > A/C > G/C > T

## Data Availability

Data sharing not applicable. No new data were created or analyzed in this study. Data sharing is not applicable to this article.

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
