# Peer review of "Associations of BDNF/BDNF-AS SNPs with Depression, Schizophrenia, and Bipolar Disorder"

_jpm, 2023, doi:10.3390/jpm13091395_

Round 1

Reviewer 1 Report

This is a thoughtful and well written review of the genetic variation within the BDNF gene.

In general, the field seems to be moving away from candidate gene approaches. However, given the role of BDNF in so many disorders,  this review could serve as an important reference for studies seeking to understand individual SNPs within larger contexts.

A number of studies have however shown that BDNF genetic variation only has impact in specific situations- e.g., in moderation with stress or other risk factors, or in specific subpopulations. A final paragraph summarizing the moderating factors would be helpful to future investigators.

N/A

Author Response

Dear Reviewer,

Thank you for your thoughtful review and kind words about our manuscript. We greatly appreciate your feedback on our paper "Associations of BDNF/BDNF-AS SNPs with Depression, Schizophrenia, and Bipolar Disorder".

This is a thoughtful and well written review of the genetic variation within the BDNF gene.

In general, the field seems to be moving away from candidate gene approaches. However, given the role of BDNF in so many disorders, this review could serve as an important reference for studies seeking to understand individual SNPs within larger contexts.

A number of studies have however shown that BDNF genetic variation only has impact in specific situations- e.g., in moderation with stress or other risk factors, or in specific subpopulations. A final paragraph summarizing the moderating factors would be helpful to future investigators.

We conducted a thorough review of numerous articles to compile a comprehensive overview of the existing literature on “Associations of BDNF/BDNF-AS SNPs with Depression, Schizophrenia, and Bipolar Disorder”. Our primary objective was to present readers with a consolidated resource that highlights key findings in this area, potentially paving the way for personalized medicine approaches in mental health disorders. Our paper is designed as a review article, focusing on summarizing the main outcomes of various studies. This approach enables readers to access critical insights from multiple sources in a single location.

The complexity of interactions between BDNF polymorphisms and multiple factors, such as “stress or other risk factors, or in specific subpopulations”, indeed plays a crucial role in understanding the nuanced effects of these genetic variations. As highlighted in our concluding paragraphs, we aimed to capture the “intricate interplay between SNPs of BDNF and BDNF-AS genetic variations, gene-gene interactions, environmental factors, and their relation to mental health disorders and treatment outcomes in diverse patient populations”. Furthermore, in our conclusion, we underscore the impact of these diverse findings, noting that “While some inconsistencies exist across studies, these discrepancies mainly arise from the natural diversity among populations and potential confounding factors, highlighting the dynamic nature of genetic influences on mental health disorders”.

Additionally, we have revised our paper and made some important additions to enhance its comprehensiveness:

  • We have expanded our discussion to include the impact of "diet" and "exercise" on BDNF.

To accommodate these additions, we made adjustments within the existing content to provide a logical continuation of the discussion. Furthermore, we have included new references (line 1537) to support the updated content. Here are the additional paragraphs (line 899):

These findings underscore the importance of considering gene-environment interactions in understanding the impact of BDNF SNPs on the development of mental health disorders. Indeed, there are other studies that reveal a complex interplay of gene-environment interactions, where the outcomes associated with specific alleles depend on the presence or absence of certain environmental influences. Moreover, the impact of alleles is not uniform across all environmental conditions. For instance, Met allele carriers with low levels of neglect display higher depressive symptoms compared to those with the Val/Val genotype. However, in cases of higher neglect, depressive symptoms increase only in individuals with the Val/Val genotype [75]. Furthermore, individuals homozygous for the Met allele show an increased predisposition to higher levels of depressive symptoms when exposed to elevated levels of childhood stress. However, interestingly, the same Met/Met genotype experiences a decrease in depressive symptoms with an increase in physical activity [79].

Indeed, physical activity has demonstrated a significant influence on BDNF. Szuhany et al. (2015) conducted a meta-analytic review, revealing that exercise can influence BDNF activity. Each exercise session contributes to a moderate increase in BDNF levels, and consistent physical activity can further amplify this effect. While regular exercise affects resting BDNF levels, its impact appears less pronounced compared to the immediate post-exercise effect and may not be considered robust. Notably, females demonstrated a less pronounced increase in BDNF levels following exercise compared to males [167]. However, another study found no increase in resting serum BDNF levels after 6 months of aerobic training, resistance training, or their combination in adolescents with overweight and obesity [168]. In contrast, healthy males aged 53 to 64 years demonstrated a reduction in BDNF serum levels following 12 weeks of aerobic and resistance training [169]. On the other hand, healthy older individuals experienced a significant immediate increase in serum BDNF levels following a 35-minute physical exercise session [170].

Moreover, participation in regular cardiovascular and coordination training was found to be associated with hippocampal total volume increase, demonstrated on MRI scans, after the 12-month intervention period in 62-79 year-old adults [171]. However, another study on cognitively healthy individuals above the age of 60 revealed that people with the Val/Val genotype and higher levels of physical activity exhibited greater hippocampus and temporal lobe volumes on MRI. In contrast, Met allele carriers with increased physical activity demonstrated smaller temporal lobe volume [172]. Furthermore, individuals homozygous for the Val allele above the age of 55 demonstrated improved episodic memory performance as a result of physical activity, when compared to both inactive Val homozygous individuals and active Met carriers [173]. These findings underscore the significance of personalized medicine in the realm of healthcare. Understanding how exercise and genetics impact BDNF levels and the brain in different people can help to develop personalized treatment plans rather than applying a one-size-fits-all approach.

Additionally, it is worth noting that diet and nutrition influence BDNF expression and the brain. For instance, lower hippocampal BDNF expression and decreased spatial working memory were seen in the sedentary group of rats on a high-fat, high-sucrose Western diet in comparison to the sedentary group of rats on a plant-based, amylose/amylopectin blend, lower-fat diet [174]. Interestingly, higher consumption of a Western-style dietary pattern consisting of higher saturated fats and refined carbohydrates was found in correlation with smaller left hippocampal volume on MRI scans in individuals aged above 60 years, in comparison to individuals on healthy dietary patterns [175].

  • We have incorporated additional sentences to explain “fractional anisotropy” and included the abbreviation “FA”. Here are these sentences (line 186):

“Val66Met BDNF polymorphism may affect white matter fiber tracts, which can be evaluated by fractional anisotropy (FA), assessing the direction and diffusion of water in brain tissue with the usage of MRI with diffusion tensor imaging (DTI). This technique helps to identify microstructural abnormalities and changes in white matter integrity [77].

3)    Upon reviewing this (line 231) paragraph, we have decided to edit it from “the quadratic cortisol function” to simply “cortisol” for better clarity, as the following sentence already explains its relationship “U-shaped relationship between the mean adjusted cortisol at baseline and depressive onsets”.

4)    We corrected a few typos, which we highlighted in yellow in the revised manuscript.

In conclusion, we would like to extend our sincere appreciation to the reviewer for the valuable comments and insightful suggestions. Your feedback has been instrumental in refining our manuscript, and we are truly grateful for your time. We believe that our meticulous work on “Associations of BDNF/BDNF-AS SNPs with Depression, Schizophrenia, and Bipolar Disorder” holds significant value for the scientific community and readers alike. We are excited about the opportunity to share our findings and contribute to the advancement of knowledge in the field.

Thank you once again for your invaluable contributions, and we look forward to the possibility of seeing our work published in this remarkable Journal of Personalized Medicine.

Warm regards,

Reviewer 2 Report

Thank you for this informative and important review. Your paper covers a mountain of research. My man concern is that it fails to impress upon the reader what can be sorted out from that mountain. Section after section, I kept writing a comment like that on SNP rs6265 and schizophrenia (or bipolar disorder):

With so many different findings across the many studies, and with most relating to a single or only a few symptoms or background factors, how can we tell what is a random finding and what is worthy of investigating further—or applying to cases? If any form of meta-analysis of these data is possible, it seems needed to make sense of all these findings.

Or, after the Discussion’s portion on cognitive function and IQ:

Some sort of meta-analysis or at least a tree plot seems needed to tell whether rs 6265 variants cause differences in cognitive function/IQ.

Even a tabulation with the main negative and positive finding in each study would help.

The problem is that there is no weighting. We can’t tell from the review what the statistical power of the linkage studies was, so we cannot tell which results to accept and consider for treatment—or whether contradictory results should cancel each other in our judgment - -and which results are of such little power that they could be productively left aside.

Another problem for such a review is the disparity between populations considered as a single disorder, e.g., childhood, young-adult, and adult BD. Particularly, I was concerned on seeing “Anxiety Disorder” as if that were a single entity. For the “Journal of Personalized Medicine,” some of the discussion and conclusions ought to say how interaction between BDNF variants and epigenetic factors including neglect, abuse, and negative life events play out in the individual case.

I am also concerned that the introduction did not tell us more about what we already know clinically about BDNF: that exercising increases BNDF, or that such is associated with better outcomes in depression. Or that a high-fat, high-sugar diet is associated with low BNDF and schizophrenia. https://pubmed.ncbi.nlm.nih.gov/15041037/

I also submit to you the few language/typo issues I found.

Line 88 diosder -> disorder

Line 101 blood have -> blood levels have

130  Would you mind adding Bipolar Disorder before your first abbreviation of it as BD?

Throughout ”Clinical Correlations,” you revert to using “BDNF.” Can we be sure this means m-BDNF? It’s a natural question to ask after the previous section, so please tell us. 

187 “fractional anisotropy” may be familiar to neuroscientists but may not be to other readers. Can you add a reference before the first one you do cite, namely a review of the importance of FA to neuronal function?

227 Please define “Quadratic cortisol function.” It’s not relevant that I as a clinician don’t know what it means. But it must be relevant to clarifying your review that the term cannot be found by google search.

Rs6265 and schizophrenia   With so many different findings across the many trials, and with most relating to a single or only a few symptoms or background factors, how can we tell what is a random finding and what is worthy of investigating further—or applying to cases? If any form of meta-analysis of these data is possible, it seems needed to make sense of all these findings.

450 “notable,” -> notably (not followed by a comma)

866-884. Some sort of meta-analysis or at least a tree plot seems needed to tell whether the genotype causes differences in cognitive function/IQ.

886 delete “nevertheless” because it contradicts what you seem to be concluding.

Author Response

Dear Reviewer,

Thank you for your constructive feedback on our paper "Associations of BDNF/BDNF-AS SNPs with Depression, Schizophrenia, and Bipolar Disorder". We appreciate your thoughtful evaluation and the opportunity to address your comments and suggestions, as outlined below:

  1. “Thank you for this informative and important review. Your paper covers a mountain of research. My man concern is that it fails to impress upon the reader what can be sorted out from that

mountain. Section after section, I kept writing a comment like that on SNP rs6265 and schizophrenia (or bipolar disorder):

With so many different findings across the many studies, and with most relating to a single or only a few symptoms or background factors, how can we tell what is a random finding and what is worthy of investigating further—or applying to cases? If any form of meta-analysis of these data is possible, it seems needed to make sense of all these findings.

Or, after the Discussion’s portion on cognitive function and IQ:

Some sort of meta-analysis or at least a tree plot seems needed to tell whether rs 6265 variants cause differences in cognitive function/IQ.

Even a tabulation with the main negative and positive finding in each study would help.

The problem is that there is no weighting. We can’t tell from the review what the statistical power of the linkage studies was, so we cannot tell which results to accept and consider for treatment—or

whether contradictory results should cancel each other in our judgment - -and which results are of such little power that they could be productively left aside.”

We conducted a thorough review of numerous articles to compile a comprehensive overview of the existing literature on BDNF/BDNF-AS gene polymorphism in relation to Depression, Schizophrenia, and Bipolar Disorder. Our primary objective was to present readers with a consolidated resource document that highlights key findings in this area, potentially paving the way for personalized medicine approaches in mental health disorders.

Our paper is designed as a review article, focusing on summarizing the main outcomes of various studies. This approach enables readers to access critical insights from multiple sources in a single location. We acknowledge that we did not undertake a meta-analysis, tree plot, or tabulation, as these would have significantly extended the article's length and complexity that possibly would overwhelm the readers. Instead, we provided references for each study cited in our paper, with the expectation that interested readers would choose to explore the original research articles for in-depth details, including statistical power analyses.

We believe that our review manuscript serves as a valuable knowledge repository for individuals seeking to delve into the topic of "Associations of BDNF/BDNF-AS SNPs with Depression, Schizophrenia, and Bipolar Disorder." It is intended to facilitate accessibility to pertinent findings and encourage further exploration into this critical area of research. We sincerely hope that the reviewer will accept our point of view.

  1. “Another problem for such a review is the disparity between populations considered as a single disorder, e.g., childhood, young-adult, and adult BD.

We agree with the reviewer comment. However we choice not to segregate studies by age group, such as “childhood, young-adult, and adult BD”, because, in our review manuscript, we organized studies based on the specific SNP associations and mental health disorders, namely Depression, Schizophrenia, and Bipolar Disorder.

  1. Particularly, I was concerned on seeing “Anxiety Disorder” as if that were a single entity.

We appreciate your attention to the mention of “Anxiety” in our manuscript and would like to clarify our approach. The main focus of our review manuscript was to explore associations between BDNF/BDNF-AS SNPs and specific mental health disorders, namely Depression, Schizophrenia, and Bipolar Disorder. Consequently, we did not plan to include “Anxiety Disorder” as a separate topic, as it was outside the scope of our review. We agree anxiety disorder deserve a separate review given the diversity of multiple conditions generally subsumed under anxiety disorders.

However, we did mention “Anxiety” in two instances within our manuscript:

  • We discussed “anxiety symptoms” in children because they were found to be linked to Bipolar Disorder in their “biological parents”. Here is this paragraph (line 422):

“According to the study by Park et al. (2015), children (aged 9-18) with the val/val genotype of biological parents diagnosed with BD displayed notably elevated levels of anxiety compared to BD offspring with val/met or met/met genotype. This finding suggests that subclinical anxiety symptoms in these children may serve as early prodromal indicators of BD and potentially early manifestations of an anxiety disorder [114].”

  • We also mentioned anxiety as a comorbidity with Bipolar Disorder, as we believed it was important for readers to understand this comorbidity in the context of the rs6265 polymorphism. Here is this paragraph (line 441):

“Chang et al. (2013) conducted a study among Han Chinese in Taiwan and found that the rs6265 Val/Val genotype of the BDNF polymorphism correlated with BD-I comorbid with anxiety disorder (AD). They also observed an interaction between the rs6265 Val/Val genotype and the Gly/Gly genotype of the DRD3 Ser9Gly polymorphism in BD-II comorbid with AD, but not in BD-II without AD comorbidity, when compared to healthy controls. DRD3 is the dopamine D3 receptor gene on chromosome 3q13.3. These findings suggest that the rs6265 polymorphism is associated with BD-I comorbid with AD and may modulate the influence of the DRD3 Ser9Gly polymorphism in BD-II comorbid with AD [119]. Interestingly, Lee et al. (2013) discovered a significant interaction effect for the Val/Val genotypes of the BDNF Val66Met polymorphism and the Val/Met and Met/Met genotypes of the COMT Val158Met polymorphism among BD-II without AD patients compared to healthy controls [120].”

  1. For the “Journal of Personalized Medicine,” some of the discussion and conclusions ought to say how interaction between BDNF variants and epigenetic factors including neglect, abuse, and negative life events play out in the individual case.”

Thank you for highlighting the “interaction between BDNF variants and epigenetic factors, including neglect, abuse, and negative life events”. We acknowledge the intricacies of this relationship and have indeed delved into these aspects within our manuscript. We also discussed this complexity as a "three-way interaction" (line 949) involving "gene-gene interactions," along with the influence of "negative life events." Furthermore, we have expanded our discussion to include the impact of "diet" and "exercise" on BDNF. Thank you for this suggestion, we believe that these additions contribute to a more comprehensive understanding of the multifaceted aspects of BDNF polymorphism interactions with various environmental factors. We hope that these additions enhance the depth of the discussion part of our manuscript.

  1. I am also concerned that the introduction did not tell us more about what we already know clinically about BDNF: that exercising increases BNDF, or that such is associated with better outcomes in depression. Or that a high-fat, high-sugar diet is associated with low BNDF and schizophrenia. https://pubmed.ncbi.nlm.nih.gov/15041037/

Thank you for sharing the link “https://pubmed.ncbi.nlm.nih.gov/15041037/” to the study by Peet M. (2004). Nutrition and schizophrenia: beyond omega-3 fatty acids. Prostaglandins, leukotrienes, and essential fatty acids, 70(4), 417–422. https://doi.org/10.1016/j.plefa.2003.12.019

We appreciate your comment about further information regarding clinical aspects of BDNF, and recommendation to discuss “exercise” and “diet” as influencing factors on BDNF. In our manuscript's discussion, we mentioned our focus on studies published from 2012 “It is important to note that the scope of our review is limited to studies published from 2012 and identified through PubMed citations.” (line 984) This choice was made to maintain consistency with the scope of our review and to ensure fairness to authors of earlier research papers. However, in response to your suggestion, we have incorporated additional paragraphs into the “discussion” section where it was relevant to discuss environmental factors influencing BDNF. To accommodate these additions, we made adjustments within the existing content to provide a logical continuation of the discussion. Furthermore, we have included new references (line 1537) to support the updated content.

Here are the new paragraphs (line 899):

These findings underscore the importance of considering gene-environment interactions in understanding the impact of BDNF SNPs on the development of mental health disorders. Indeed, there are other studies that reveal a complex interplay of gene-environment interactions, where the outcomes associated with specific alleles depend on the presence or absence of certain environmental influences. Moreover, the impact of alleles is not uniform across all environmental conditions. For instance, Met allele carriers with low levels of neglect display higher depressive symptoms compared to those with the Val/Val genotype. However, in cases of higher neglect, depressive symptoms increase only in individuals with the Val/Val genotype [75]. Furthermore, individuals homozygous for the Met allele show an increased predisposition to higher levels of depressive symptoms when exposed to elevated levels of childhood stress. However, interestingly, the same Met/Met genotype experiences a decrease in depressive symptoms with an increase in physical activity [79].

Indeed, physical activity has demonstrated a significant influence on BDNF. Szuhany et al. (2015) conducted a meta-analytic review, revealing that exercise can influence BDNF activity. Each exercise session contributes to a moderate increase in BDNF levels, and consistent physical activity can further amplify this effect. While regular exercise affects resting BDNF levels, its impact appears less pronounced compared to the immediate post-exercise effect and may not be considered robust. Notably, females demonstrated a less pronounced increase in BDNF levels following exercise compared to males [167]. However, another study found no increase in resting serum BDNF levels after 6 months of aerobic training, resistance training, or their combination in adolescents with overweight and obesity [168]. In contrast, healthy males aged 53 to 64 years demonstrated a reduction in BDNF serum levels following 12 weeks of aerobic and resistance training [169]. On the other hand, healthy older individuals experienced a significant immediate increase in serum BDNF levels following a 35-minute physical exercise session [170].

Moreover, participation in regular cardiovascular and coordination training was found to be associated with hippocampal total volume increase, demonstrated on MRI scans, after the 12-month intervention period in 62-79 year-old adults [171]. However, another study on cognitively healthy individuals above the age of 60 revealed that people with the Val/Val genotype and higher levels of physical activity exhibited greater hippocampus and temporal lobe volumes on MRI. In contrast, Met allele carriers with increased physical activity demonstrated smaller temporal lobe volume [172]. Furthermore, individuals homozygous for the Val allele above the age of 55 demonstrated improved episodic memory performance as a result of physical activity, when compared to both inactive Val homozygous individuals and active Met carriers [173]. These findings underscore the significance of personalized medicine in the realm of healthcare. Understanding how exercise and genetics impact BDNF levels and the brain in different people can help to develop personalized treatment plans rather than applying a one-size-fits-all approach.

Additionally, it is worth noting that diet and nutrition influence BDNF expression and the brain. For instance, lower hippocampal BDNF expression and decreased spatial working memory were seen in the sedentary group of rats on a high-fat, high-sucrose Western diet in comparison to the sedentary group of rats on a plant-based, amylose/amylopectin blend, lower-fat diet [174]. Interestingly, higher consumption of a Western-style dietary pattern consisting of higher saturated fats and refined carbohydrates was found in correlation with smaller left hippocampal volume on MRI scans in individuals aged above 60 years, in comparison to individuals on healthy dietary patterns [175].

  1. I also submit to you the few language/typo issues I found.

Line 88 diosder -> disorder

Thank you for pointing out this typo. We corrected it to "disorder”.

  1. Line 101 blood have -> blood levels have

Thank you, we corrected it to “blood levels have“.

  1. 130 Would you mind adding Bipolar Disorder before your first abbreviation of it as BD?

Yes, we have included “bipolar disorder (BD)” in the line under the number “90” in our manuscript.

  1. Throughout ”Clinical Correlations,” you revert to using “BDNF.” Can we be sure this means m-BDNF? It’s a natural question to ask after the previous section, so please tell us.

Yes, we appreciate your question and understand the need for clarification regarding the interchangeable usage of "m-BDNF" and "BDNF" terms. To address this, we have included the following sentence (line 41) in our manuscript: “Then, pro-BDNF is cleaved intracellularly into the mature form (m-BDNF), which is then released into the extracellular space and is simply called BDNF [12-17].”

  1. 187 “fractional anisotropy” may be familiar to neuroscientists but may not be to other readers. Can you add a reference before the first one you do cite, namely a review of the importance of FA to neuronal function?

Yes, we greatly appreciate your suggestion. In response, we have incorporated additional sentences to explain “fractional anisotropy” and included the abbreviation “FA”.

Here is the existing paragraph (line 186):

“Val66Met BDNF polymorphism may affect white matter fiber tracts, which can be evaluated by fractional anisotropy (FA), assessing the direction and diffusion of water in brain tissue with the usage of MRI with diffusion tensor imaging (DTI). This technique helps to identify microstructural abnormalities and changes in white matter integrity [77]. According to Carballedo et al. (2012), individuals with MDD who possess the Met allele of the BDNF gene exhibit reduced FA in the uncinate fasciculus compared to MDD patients who are homozygous for the Val allele and healthy individuals carrying the Met allele. The study also found a significant three-way interaction involving the cingulum (dorsal, rostral, and parahippocampal regions), brain hemisphere, and BDNF genotype. Specifically, Met allele carriers demonstrated higher FA in the left rostral cingulum than Val/Val allele carriers. This suggests that the Met allele of the BDNF polymorphism may increase susceptibility to dysfunctions associated with the uncinate fasciculus, which is known to be involved in negative emotional-cognitive processing bias, autonoetic self-consciousness, and declarative memory issues. The findings highlight the significance of neurotrophic involvement in the connections between the limbic system and the prefrontal cortex [77].”

  1. 227 Please define “Quadratic cortisol function.” It’s not relevant that I as a clinician don’t know what it means. But it must be relevant to clarifying your review that the term cannot be found by google search.

Thank you for bringing up the term “Quadratic cortisol function”. We used this term for continuity in reference to the study mentioned. To clarify, a “quadratic function” refers to the graph of a U-shaped curve known as a parabola (source: https://www.andrews.edu/~rwright/Precalculus-RLW/Text/02-02.html).

Upon reviewing this paragraph, we have decided to edit it (line 231) from “the quadratic cortisol function” to simply “cortisol” for better clarity, as the following sentence already explains its relationship “U-shaped relationship between the mean adjusted cortisol at baseline and depressive onsets”.

  1. Rs6265 and schizophrenia With so many different findings across the many trials, and with most relating to a single or only a few symptoms or background factors, how can we tell what is a random finding and what is worthy of investigating further—or applying to cases? If any form of meta-analysis of these data is possible, it seems needed to make sense of all these findings.

We appreciate the reviewer comment. However, the extensive research on the association between rs6265 and schizophrenia prompted us to compile a comprehensive overview of the various associations reported in studies conducted from 2012 to the present. While meta-analysis is undoubtedly a valuable method for consolidating findings, it is important to note that this manuscript primarily serves as a review article. Therefore, we believe that conducting a meta-analysis would be better suited for a distinct research endeavor and a subsequent publication.

  1. 450 “notable,” -> notably (not followed by a comma)

Thank you, we corrected it to “notably”.

  1. 866-884. Some sort of meta-analysis or at least a tree plot seems needed to tell whether the genotype causes differences in cognitive function/IQ.

In principal, we agree with the reviewer that meta-analysis informative. However, in this review manuscript, our primary goal was to present readers with a comprehensive overview of the existing findings. While the suggestion of a "meta-analysis" or "a tree plot" specifically focused on "cognitive function/IQ" is valuable, we think that it could be a promising avenue for future research and a separate article dedicated to this specific topic.

  1. 886 delete “nevertheless” because it contradicts what you seem to be concluding.

Thank you, we did remove “nevertheless”.

In conclusion, we would like to extend our sincere appreciation to the reviewer for the valuable comments and insightful suggestions. Your feedback has been instrumental in refining our manuscript, and we are truly grateful for your time. We believe that our meticulous work on “Associations of BDNF/BDNF-AS SNPs with Depression, Schizophrenia, and Bipolar Disorder” holds significant value for the scientific community and readers alike. We are excited about the opportunity to share our findings and contribute to the advancement of knowledge in the field.

Thank you once again for your invaluable contributions, and we look forward to the possibility of seeing our work published in this remarkable Journal of Personalized Medicine.

Warm regards,

Round 2

Reviewer 2 Report

No more suggestions. Thank you for clarifying the paper.